# Heterovalent-doping-enabled atom-displacement fluctuation leads to ultrahigh energy-storage density in AgNbO$_3$-based multilayer capacitors

Li-Feng Zhu[1,6], Shiqing Deng [2] ✉, Lei Zhao[3,6], Gen Li[4], Qi Wang[1], Linhai Li[1], Yongke Yan[5] ✉, He Qi[2], Bo-Ping Zhang[1] ✉, Jun Chen [2] & Jing-Feng Li [4] ✉

Dielectric capacitors with high energy storage performance are highly desired for next-generation advanced high/pulsed power capacitors that demand miniaturization and integration. However, the poor energy-storage density that results from the low breakdown strength, has been the major challenge for practical applications of dielectric capacitors. Herein, we propose a heterovalent-doping-enabled atom-displacement fluctuation strategy for the design of low-atom-displacements regions in the antiferroelectric matrix to achieve the increase in breakdown strength and enhancement of the energy-storage density for AgNbO$_3$-based multilayer capacitors. An ultrahigh breakdown strength ~1450 kV·cm$^{-1}$ is realized in the Sm$_{0.05}$Ag$_{0.85}$Nb$_{0.7}$Ta$_{0.3}$O$_3$ multilayer capacitors, especially with an ultrahigh U$_{rec}$ ~14 J·cm$^{-3}$, excellent η ~ 85% and P$_{D,max}$ ~ 102.84 MW·cm$^{-3}$, manifesting a breakthrough in the comprehensive energy storage performance for lead-free antiferroelectric capacitors. This work offers a good paradigm for improving the energy storage properties of antiferroelectric multilayer capacitors to meet the demanding requirements of advanced energy storage applications.

Dielectric capacitors with many key advantages, including high-power density, good fatigue resistance, fast charge/discharge rates, and temperature stability, are fascinatingly attractive for applications in pulsed-discharge and power conditioning for electronic systems, such as space shuttle power systems, high-powered accelerators, hybrid electric vehicles, and kinetic energy weapons[1–6]. However, its energy-storage density, being far lower than that of batteries or electrochemical capacitors, is still too inferior to satisfy application requirements in these areas. Therefore, extensive endeavors have been continuously devoted to improving their energy densities to meet the demands of integration, compactness, and miniaturization of electronic devices[7–9].

Various types of dielectric materials can be potential candidates for energy storage, including antiferroelectrics (AFEs)[10–12], relaxor ferroelectrics (RFEs)[13,14], normal ferroelectrics (FEs)[15], and linear nonpolar dielectric materials[16]. Among these dielectrics, AFE dielectrics, characterized by a double hysteresis loop, are favored for energy storage due to their relatively high maximum polarization ($P_{max}$) and

[1]School of Materials Science and Engineering, University of Science and Technology Beijing, Beijing 100083, China. [2]Beijing Advanced Innovation Center for Materials Genome Engineering, University of Science and Technology Beijing, Beijing 100083, China. [3]College Physics Science & Technology, Hebei University, Baoding 071002, China. [4]State Key Laboratory of New Ceramics and Fine Processing, School of Materials Science and Engineering, Tsinghua University, Beijing 100084, China. [5]Electronic Materials Research Laboratory, Key Laboratory of the Ministry of Education & International Center for Dielectric Research, School of Electronic Science and Engineering, Xi'an Jiaotong University, Xi'an 710049 PR, China. [6]These authors contributed equally: Li-Feng Zhu, Lei Zhao. ✉e-mail: sqdeng@ustb.edu.cn; yanyongke@xjtu.edu.cn; bpzhang@ustb.edu.cn; jingfeng@mail.tsinghua.edu.cn

particularly low remanent polarization ($P_r$) compared with other types of dielectrics. In the past decades, lead-based AFE materials that possess excellent recoverable energy-storage density ($U_{rec}$) and efficiency ($\eta$), like (Pb,La)(Zr,Ti)O$_3$ system[10,11,17–19], have been the mainstay energy storage materials. While increasing environmental concerns necessitate the development of lead-free AFE energy storage ceramics[20–25]. As a representative AFE system, the AgNbO$_3$ (AN) shows great potential for energy storage due to its large polarization up to 52 $\mu$C cm$^{-2}$ and has been at the focal plane of the research since the first discovery of its AFE characteristics[23]. However, the $U_{rec}$ for pure AN ceramic is only about 2 J cm$^{-3}$[24,25], which is far lower than that of PbZrO$_3$-based AFE systems[17–19]. This essentially results from the non-zero $P_r$ and weak breakdown strength (BDS) at room temperature of pure AN ceramic. Researchers deem that the non-zero $P_r$ characteristic of AN ceramic is related to its phase structure, which is not the centrosymmetric Pbcm structure (AFE phase), but the orthorhombic Pmc2$_1$ structure with uncompensated ion displacement (FIE phase). Thus, improving the energy storage performance of AN based on the simultaneous decrease in $P_r$ and increase in BDS remains a challenging task, particularly considering the still ambiguous phase structure.

It is well recognized that large $P_{max}$ in the field-induced ferroelectric phase and zero $P_r$, as well as high BDS in the AFE phase, are desired to achieve high energy-storage density. To meet these criteria, a series of strategies have been developed to reduce $P_r$ and enhance BDS for the AN system, which is mainly from the following three aspects. One is using oxide dopants for compositional modification to suppress the ferroelectricity and boost the antiferroelectricity at room temperature. Typical examples include AN + 0.1 wt% MnO$_2$[26] and AN + 0.1 wt% WO$_3$[27] systems, where the $U_{rec}$ reaches 2.5 and 3.3 J cm$^{-3}$, respectively. Another is ion substitutions, e.g., replacement of Ag$^+$ by La$^{3+}$[28], Sm$^{3+}$[29], Ba$^{2+}$[30], Lu$^{3+}$[31], Gd$^{3+}$[32], etc., and/or Nb$^{5+}$ by Ta$^{5+}$[33]. The $U_{rec}$ can be effectively increased to 3.2 J cm$^{-3}$ in Ag$_{1-x}$La$_x$NbO$_3$ system at $x = 0.02$[28], 4.5 J cm$^{-3}$ in the Ag$_{1-3x}$Sm$_x$NbO$_3$ system at $x = 0.02$[29], 2.3 J cm$^{-3}$ in Ag$_{1-2x}$Ba$_x$NbO$_3$ system at $x = 0.02$[30], and 4.2 J cm$^{-3}$ in Ag(Nb$_{1-x}$Ta$_x$)O$_3$ system at $x = 0.15$[33], and so on. Reducing the thickness of the dielectric layer is the other efficacious strategy to enhance the BDS and $U_{rec}$ of the AN system. For example, multilayer capacitors (MLCCs) can possess a thin dielectric layer down to 10–20 $\mu$m in thickness. The BDS of MLCCs prepared by tape casting can be enhanced more than 3 times in comparison to the monolithic ceramic capacitor[34]. For instance, for 0.61BiFeO$_3$–0.33(Ba$_{0.8}$Sr$_{0.2}$)TiO$_3$–0.06La(Mg$_{2/3}$Nb$_{1/3}$)O$_3$ MLCCs system, the BDS is up to 730 kV cm$^{-1}$, which is far higher than 230 kV cm$^{-1}$ for ceramic bulks. The $U_{rec}$ of MLCCs is about 3 times as high as ceramic bulks. Similarly, high BDS (over 700 kV/cm) and excellent $U_{rec} = 10.5$ J cm$^{-3}$ also have been achieved in (0.7−$x$)BiFeO$_3$–0.3BaTiO$_3$–$x$Nd(Zn$_{0.5}$Zr$_{0.5}$)O$_3$ MLCCs[35]. Our previous work achieved an ultrahigh BDS -1020 kV cm$^{-1}$ in Ag(Nb$_{0.85}$Ta$_{0.15}$)O$_3$ + 0.25 wt% MnO$_2$ multilayer capacitors, which can exhibit an excellent $U_{rec} = 7.9$ J cm$^{-3}$ and $\eta = 71\%$[36]. All these results show that the AN system has great potential for energy storage application, and, more importantly, suggest the preparation of MLCCs as the most effective strategy to boost the BDS and $U_{rec}$ of the AN system. Unfortunately, a large electric-field-induced internal stress due to the AFE-to-FE phase transformation strain can appear in the AFE MLCCs devices, which greatly hinders the further improvement of the BDS and $U_{rec}$. Although the design of the ⟨111⟩ textured NBT-SBT MLCCs reported by Li et al.[21] has been demonstrated to be effective in reducing such internal stress, its general applicability in other AFE systems remains elusive.

In this work, a heterovalent-doping-enabled atom-displacement fluctuation strategy is proposed for the design of low-atom-displacements (LAD) regions in the AFE matrix to effectively reduce the electric-field-induced strain to improve the energy storage performance in AN-based systems. This is achieved by the respective substitution of heterovalent rare earth ions (RE$^{3+}$) and isovalent Ta$^{5+}$ for

Ag$^+$ and Nb$^{5+}$ ions. As a result, an ultrahigh $U_{rec}$ -14 J cm$^{-3}$ and excellent $\eta$ ~85% have been realized in the Sm$_{0.05}$Ag$_{0.85}$Nb$_{0.7}$Ta$_{0.3}$O$_3$ multilayer capacitors, which characterize an ultrahigh breakdown strength of -1450 kV cm$^{-1}$ and $P_{D,max}$ ~102.84 MW cm$^{-3}$. Atomic-scale electron microscopy studies reveal the complicated local structures and highlight the critical roles of the LAD region in boosting energy storage properties. Our developed new strategy of the heterovalent-doping can be generally applicable to numerous antiferroelectrics, as well as ferroelectrics and dielectrics, which would navigate the discovery and development of superior energy storage materials.

The design idea of the material is illustrated in Fig. 1. It is well known that the phase structure of AN samples is ferroelectric (FIE) Pmc2$_1$ phase, whose cations show displacement along the $\pm[1\bar{1}0]_C$ direction, forming a periodic variation along the c-axis direction as shown in Fig. 1a. Because of the asymmetric atomic displacement arrangement, an uncompensated polarization configuration is generated, for example, $P_S^+ + P_S^- > 0$ as shown in Fig. 1a[37,38]. However, when a small amount of Nb$^{5+}$ is replaced by the Ta$^{5+}$ ions, the phase structure of Ag(Nb$_{1-x}$Ta$_x$)O$_3$ samples turns into an AFE phase[39], in which the $P_S^+ + P_S^-$ is zero as shown in Fig. S1. Moreover, two V$'_{!Ag^+}$ defects or one V$'_{!Ag^+}$ - RE$^{3+}$-V$'_{!Ag^+}$ defect dipole will be generated in (RE$_x$Ag$_{1-3x}$)(Nb,Ta)O$_3$ system when the Ag$^+$ ions are replaced by the heterovalent rare earth RE$^{3+}$ ion. In this scenario, the periodic variation of atom-displacement fluctuation in the AFE phase would be destroyed, and the Ag$^+$ ions near the V$'_{!Ag^+}$ defects can migrate more closely to V$'_{Ag^+}$ defects, forming LAD regions in the (RE$_x$Ag$_{1-3x}$)(Nb,Ta)O$_3$ system as shown in Fig. 1b. Figure 1c, d illustrates the diagram of phase transition and volume expansion process of AN and (RE$_x$Ag$_{1-3x}$)(Nb,Ta)O$_3$ capacitors as a high electric field $E > E_F$ is applied. Because of the appearance of the LAD region, which belongs to the low polarization region, the degree of volume change caused by the AFE-FE phase transition for (RE$_x$Ag$_{1-3x}$)(Nb,Ta)O$_3$ samples is far lower than that of the AN sample. Figure 1e compares the electric-field-induced strain of AN, ANT, and (Sm$_{0.05}$Ag$_{0.85}$)(Nb$_{0.70}$Ta$_{0.30}$)O$_3$ (SANT) ceramics. As expected, SANT ceramic, with the appearance of the weak-polarization LAD region, possesses very low electric-field-induced strain compared with those of AN and ANT capacitors. Figure S2 shows the finite-element simulations for the strain distribution of AN-based, ANT-based, and SANT-based MLCCs at different external electric fields ($E$). When the electric field ($E$) is below $E_F$, all samples exhibit a low electric-field-induced internal stress. However, when $E$ is larger than $E_F$, the AN-based and ANT-based MLCCs both show very high electric-field-induced internal stress, which is far higher than that of SANT-based capacitors. This originates from the high electric-field-induced strain in the AN-based and ANT-based capacitors, far larger than that of SANT-based capacitors, as shown in Fig. 1e. Since the low electric-field-induced internal stress is beneficial to the improvement of BDS[21], an ultrahigh BDS of about 1450 kV cm$^{-1}$ is realized in the SANT MLCCs, which is much higher than that of AN MLCCs (- 450 kV cm$^{-1}$) and ANT MLCCs (-900 kV cm$^{-1}$). Figure 1f presents a comparison of BDS and $U_{rec}$ for AN-based ceramics and MLCCs. Due to its AFE characteristic and the appearance of LAD regions, SANT MLCCs possess an ultrahigh BDS -1450 kV cm$^{-1}$ and excellent $U_{rec}$ -14 J cm$^{-3}$. These achieved BDS and $U_{rec}$ are also the highest among AN systems, as reported so far. In this sense, this work not only provides a feasible and generally applicable strategy for improving the energy storage properties of AFE MLCCs but also develops the SANT MLCCs with application potential in high-power pulse energy storage devices.

## Results

To clarify the effect of Sm and Ta co-doping on the microstructures of the AN system, we conducted atomic-scale investigations using aberration-corrected scanning transmission electron microscopy (STEM). Atomically resolved high-angle annular dark-field (HAADF) images for AN (Fig. 2a), ANT (Fig. S3), and SANT (Fig. 2d) samples at

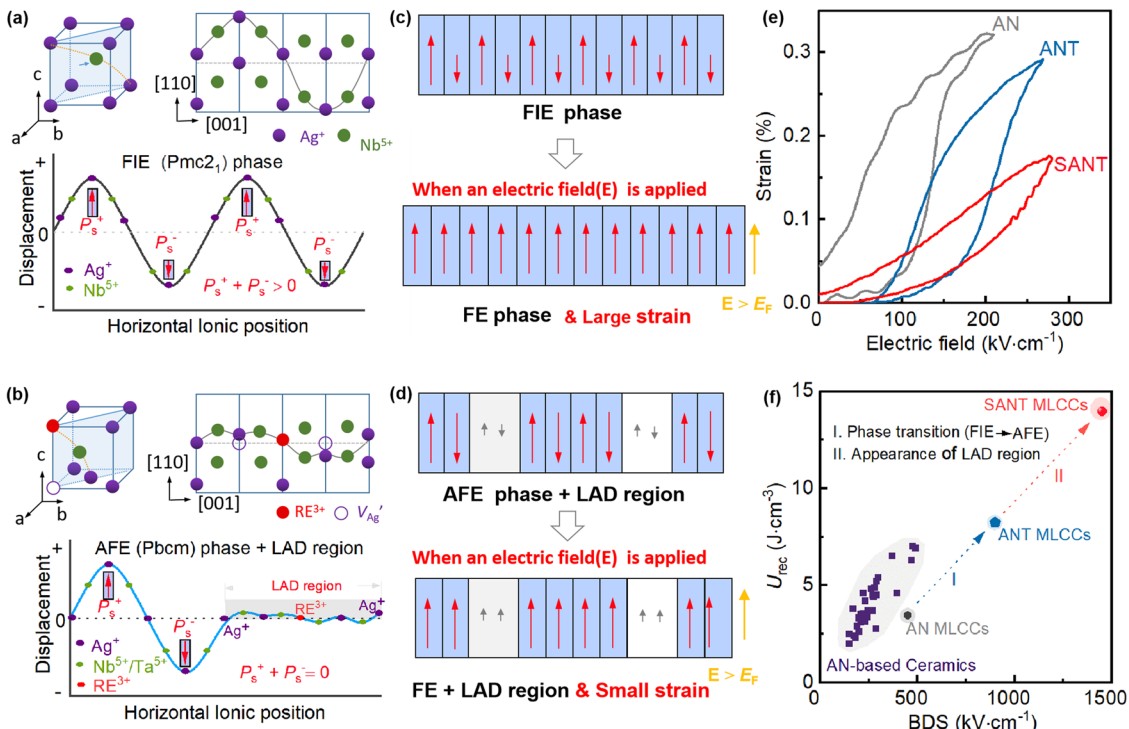

**Fig. 1 | Methods for improving the BDS and energy storage performances.** The crystal structure of pure AN corresponds to the ferroelectric Pmc2₁ phase, with cations showing displacements along the direction, forming a periodic variation along the *c*-axis direction (**a**). The crystal structure for (REₓAg₁₋₃ₓ)(Nb,Ta)O₃ system, where the periodic variation of atom-displacement fluctuation in AFE Pbcm phase is destroyed by the heterovalent rare earth ion RE³⁺ ions, forming some low-atom-displacements (LAD) region (**b**). The diagram of phase transition and volume expansion process of AN (**c**) and (REₓAg₁₋₃ₓ)(Nb,Ta)O₃ (**d**) capacitors as a high electric field $E > E_F$ is applied. The electric-field-induced strain of AN, ANT, and SANT ceramics (**e**). The comparison of BDS and $U_{rec}$ for AN-based ceramics, AN MLCCs, and SANT MLCCs (**f**).

the pseudo-cubic $[1\bar{1}0]_C$ zone axis were acquired, and quantitatively analyzed[40]. Based on the fitted atomic column positions in HAADF images, cation displacements are mapped out. cation displacements are mapped out. Figure 2b, e demonstrates atomic displacements of Ag and Nb atoms along the $\pm[1\bar{1}0]_C$ direction for AN and SANT samples, respectively. Corresponding profiles of average displacements of each vertical atomic plane are shown in Fig. 2c, f. It can be seen that for the AN sample, the cation displacements manifest a well-defined periodicity along the $[001]_C$ direction. Every successive four Ag–Nb pairs form a repeating unit, indicating the periodicity of eight atoms. The displacement magnitude is about 13 pm, as shown in Fig. 2b, c. Noteworthily, the positive and negative cation displacements are not identical, and the Ag₁ ion is above the zero line, as shown in Fig. 2c. Such characteristics suggest that the AN sample belongs to the FIE Pmc2₁ symmetry but not the AFE Pbcm symmetry, which is consistent with previous studies[38,41]. For the ANT system, the cation displacements also show analogous periodic variations along the $\pm[1\bar{1}0]_C$ direction with a slightly decreased magnitude of -10 pm, as shown in Fig. S3. While being different from that of the AN sample, the positive and negative cation displacements are almost identical as shown in Fig. S3, suggesting the AFE phase Pbcm of the ANT system.

For SANT samples, markedly different characteristics from those of AN and ANT systems are observed (Fig. 2e, f). Wherein, the periodic variations of the cation displacement are destroyed with the formation of less ordered regions, like the so-called cation periodic variation (CPV) region and LAD region. For the CPV region, although each periodicity still consists of four Ag–Nb pairs (or eight cations), like that of the AN system, the displacement magnitudes for different periodic cycles are different. Besides, the displacement amplitude for the same condition cation at different periodic for the SANT system cycles also can be different. For example, for the Ag₁ ions located at the horizontal

positions of 9, 17, and 25, their displacements along the $\pm[1\bar{1}0]_C$ direction can be positive, zero, and negative, respectively, as shown in Fig. 2f. These results indicate that the phase structure of CPV region is not the single FIE Pmc2₁ or AFE Pbcm phase, but more like the coexistence of them, which consist with the fact of the non-zero $P_r$ value of SANT MLCCs. For the LAD region, both the cation periodicity and the displacement magnitudes show an obvious disorder. For example, in the marked LAD region in Fig. 2e, the cation displacement magnitude varies in a range of between −6 pm and 0 pm. In this sense, the phase structure of the SANT sample is more complicated than the single FIE Pmc2₁ or AFE Pbcm phase, nor a simple coexistence of them. The formation mechanism of the LAD region can be understood based on the fact that the electrostatic attraction between silver vacancies and their adjacent cations can significantly distort the surrounding lattice, as shown in Fig. 1b. Due to the low cation displacement, the LAD region can be considered as a weak polarization region, which should be the essential origin for the low electric-field-induced strain in SANT samples shown in Fig. 1e.

Figure 3 presents the optical photo and scanning electron microscopy (SEM) images of the SANT multilayer capacitor, whose length × width × height is 6.0 × 4.5 × 0.5 mm. All samples characterize a high density and homogeneous grain size, which is less than 5 μm. The thickness of MLCC dielectric layers for the ANT and SANT samples is about 9.5 and 10 μm, respectively, and the Pt electrode is about 2 μm, as shown in Fig. S4. The energy dispersive spectroscopy (EDS) maps of the SANT multilayer capacitor are shown in Fig. 3d–f, which illustrates the homogeneous distribution of Ag and O elements across the capacitors, and no segregation of the elements can be observed in SANT multilayer capacitors.

The temperature-dependent dielectric permittivity and loss for the AN, ANT, and SANT MLCCs specimens are shown in Fig. 4a–c. Several typical dielectric anomaly peaks corresponding to various

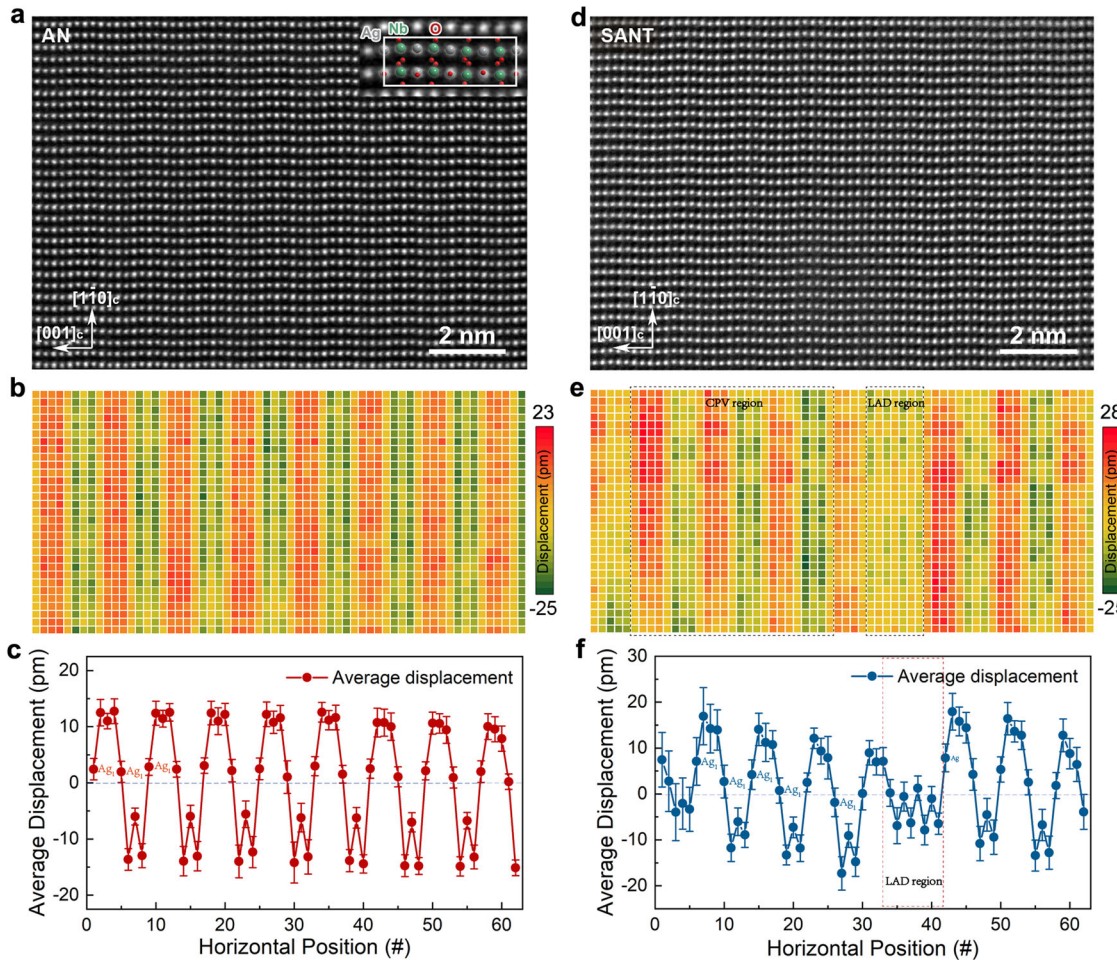

**Fig. 2 | Comparison of the atomic-scale structures of AN and SANT.** HAADF-STEM image of pure AN. Inset shows the local enlargement with the crystal structure overlaid (**a**). Map of the atomic displacements of Ag and Nb atoms along the $\pm[1\bar{1}0]_c$ direction, showing a good c-axial periodicity (**b**). Average displacement of each vertical atomic plane (**c**). HAADF-STEM image of SANT (**d**). Map of the atomic displacements of Ag, Nb, and Ta atoms along the $[1\text{-}10]_C$ direction (**e**). Both the periodicity and the displacement magnitude manifest large variations compared to that of the AN system. Two different kinds of cation displacement variation regions can be defined in the SANT system. One is the cation periodic variation (CPV) region, the other is the low-atom-displacements (LAD) region. Average displacement of each vertical atomic plane. Error bars are standard deviation (**f**).

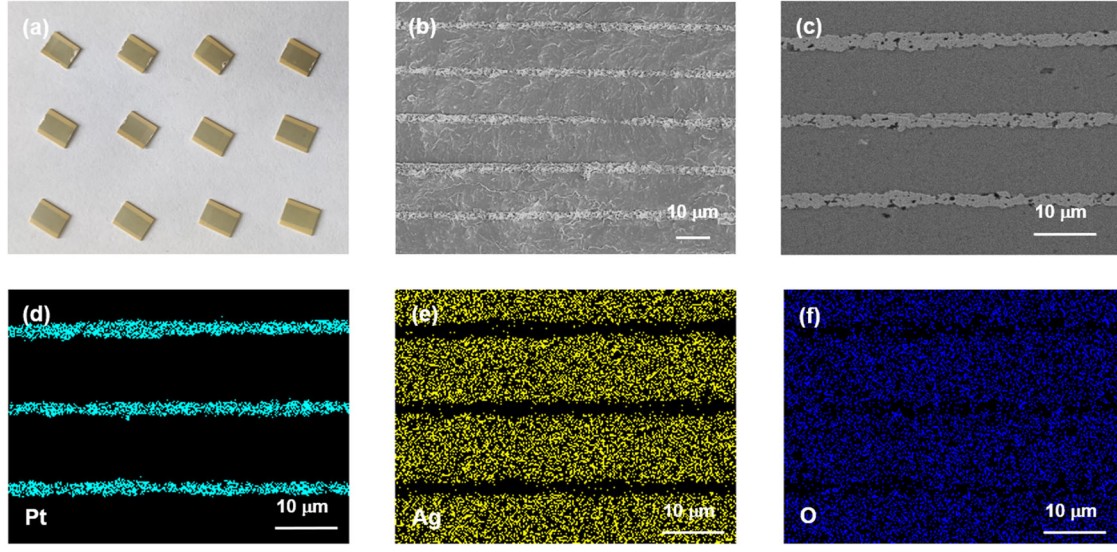

**Fig. 3 | Microstructures of the SANT multilayer capacitors.** Optical photo of the as-prepared SANT multilayer capacitors (size of the capacitor: length × width × height = 6.0 × 4.5 × 0.5 mm) (**a**). Cross-sectional SEM image (**b**), backscattered electron image (**c**), and EDS-SEM images of the elemental distributions Pt (**d**), Ag (**e**), and O (**f**) for SANT multilayer capacitors.

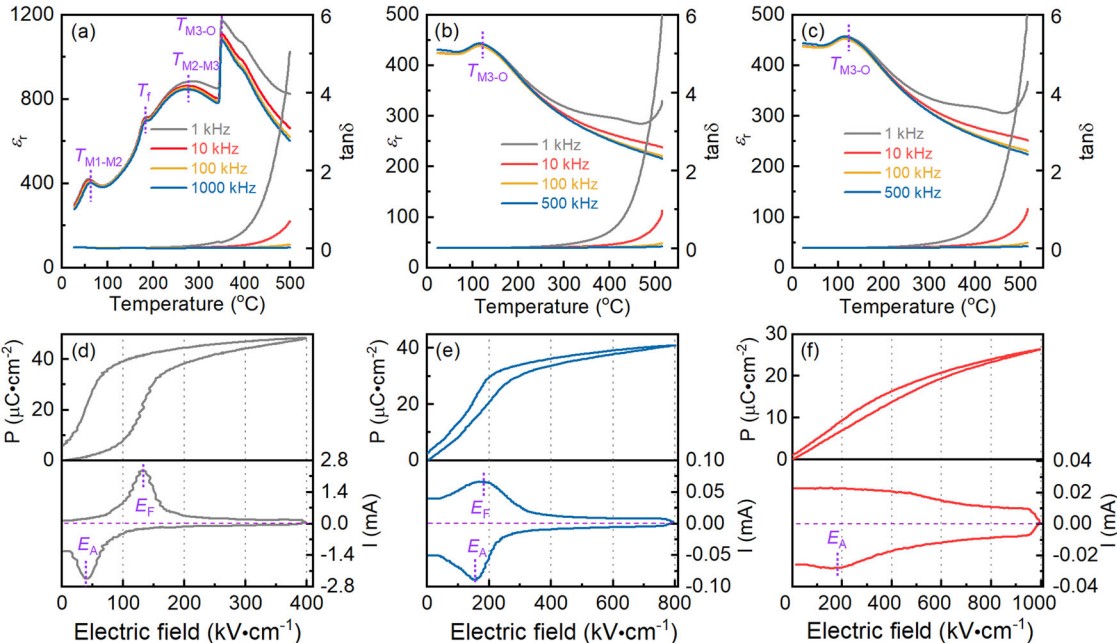

**Fig. 4 | Dielectric permittivity, unipolar P–E loops, and current curves of AN, ANT, and SANT samples.** Temperature- and frequency-dependent dielectric permittivity and loss for AN MLCCs (**a**), ANT MLCCs (**b**), and SANT MLCCs (**c**), and unipolar P–E loops and current curve for the samples of AN MLCCs (**d**), ANT MLCCs (**e**), and SANT MLCCs (**f**).

phase transitions are detected in the AN specimen, as shown in Fig. 4a, which is consistent with previous reports[33]. However, being different from AN ceramic, only one obviously dielectric anomaly peak associated with $M_3$–O phase transition is observed in ANT and SANT specimens, as shown in Figs. 4b and 4c, suggesting that the phase structure of ANT and SANT samples at room temperature is AFE $M_3$ phase. This result is different from the HAADF-STEM results, which could come from the difference between the macroscopic evaluation and microscopic characterization. The dielectric curves of ANT and SANT specimens have no obvious difference, suggesting the introduction of $Sm^{3+}$ has no effect on the point of phase transition. Figure 4d–f presents the unipolar P–E loops and current curves for the AN, ANT, and SANT MLCCs specimens. AN and ANT samples characterize a typical AFE loop, as shown in Fig. 4d, e, where two obvious current peaks correspond to $E_F$ and $E_A$ caused by the AFE-FE phase transition. However, for the SANT MLCCs specimens, its AFE characteristic is obscure, as shown in Fig. 4f, and only one current peak at $E_A$ corresponding to phase transitions from FE to AFE is detected. The difference in P–E loops between ANT and SANT MLCCs samples is unintelligible if only considering their macroscopic characteristics, as indicated by dielectric curves in Fig. 4b, c, and XRD in Fig. S5. In this sense, the discrepancy of their P–E loop should essentially derive from their different atomic-scale local structures, like differences in atom-displacement fluctuation. The disappearance of the current peak at $E_F$ for the SANT system could come from the appearance of the LAD region, which flattens the AFE to FE phase transition peak.

Figure 5a shows the BDS of AN, ANT, and SANT MLCCs. The Weibull modulus $\beta$ of all the samples is higher than 10, suggesting that the results are reliable. The BDS of AN, ANT, and SANT MLCCs is about 450, 900, and 1450 kV cm$^{-1}$, respectively. The ultrahigh BDS for SANT MLCCs is due to the reduced electric-field-induced internal stress caused by the appearance of the LAD region, as shown in Fig. 2 and Fig. S1. Figures S6–S8 exhibit the unipolar P–E loops of AN, ANT, and SANT MLCCs with the different electric fields applied. Compared with AN MLCCs, both ANT and SANT samples possess a slim P–E loop, as shown in Figs. S7 and S8. In particular, the P–E loop of the SANT system is extremely slim, indicating the high efficiency for energy storage.

The detailed variations of $P_{max}$ and $P_r$ for AN, ANT, and SANT MLCCs are displayed in Fig. 5b, c. SANT MLCCs have a lower $P_{max}$ value than AN and ANT MLCCs, as shown in Fig. 5b. This is because of the fact that the LAD region in the SANT system belongs to a weak polarization. Figure 5c shows the variation of $P_r$ value at the different electric fields. AN sample has the largest $P_r$ value, which originates from its FIE Pmc2$_1$ phase structure. The SANT sample possesses the lowest $P_r$ value. The non-zero $P_r$ values observed in ANT and SANT samples also suggest that the phase structures of ANT and SANT samples are not an ideal AFE *Pbcm* phase, but the coexistence of AFE *Pbcm* and FIE *Pmc*2$_1$ phases. This is consistent with microscopic observations as shown in Fig. 2. Figure 5d–f depicts the detailed variations of $U_{total}$, $U_{rec}$, and $\eta$ for SAN, ANT, and SANT MLCCs. Due to its high BDS, particularly low $P_r$ value and slim P–E loops, an ultrahigh $U_{rec}$ ~14 J cm$^{-3}$ and $\eta$ ~85% are achieved in SANT MLCCs, which are so far the highest values in AN system[36].

Reliable temperature stability is one of the key factors to enable the operation of devices in a wide temperature range. The temperature dependence of unipolar P–E loops of SANT MLCCs is presented in Fig. 6a, whose electric field and frequency are fixed at 800 kV cm$^{-1}$ and 10 Hz, respectively. When the temperature is between 21 °C and 60 °C, the P–E loops have little change, while the $P_{max}$, $P_r$, and hysteresis areas show a slight increase as the temperature further increases from 60 °C to 120 °C as shown in Fig. 6a. The detailed variations for $P_{max}$ and $P_r$ versus the temperature are shown in Fig. 6b. The $P_{max}$ and $P_r$ values are 24.49 μC cm$^{-2}$ and 0.55 μC cm$^{-2}$ at 21 °C, respectively. As the temperature increases to 120 °C, the $P_{max}$ and $P_r$ values significantly increase to 26.89 μC cm$^{-2}$ and 4.74 μC cm$^{-2}$. This is because of the fact that the domain orientation is enhanced since the lattice vibration is intensified as the temperature increases[42]. Nevertheless, $U_{rec}$ still shows a low variation range of less than 5% in SANT MLCCs, as shown in Fig. 6c. The frequency dependence of unipolar P–E loops is exhibited in Fig. 6d. Differently, the frequency does not affect P–E loops. The variations of $P_{max}$ and $P_r$ are displayed in Fig. 6e. The $P_{max}$ shows a slight decrease with increasing the measured frequency, which decreases from 26.18 μC cm$^{-2}$ at 1 Hz to 24.64 μC cm$^{-2}$ at 100 Hz, while the $P_r$ almost remains unchanged as the measured frequency increases. The variation range of $U_{rec}$ for SANT MLCCs with different

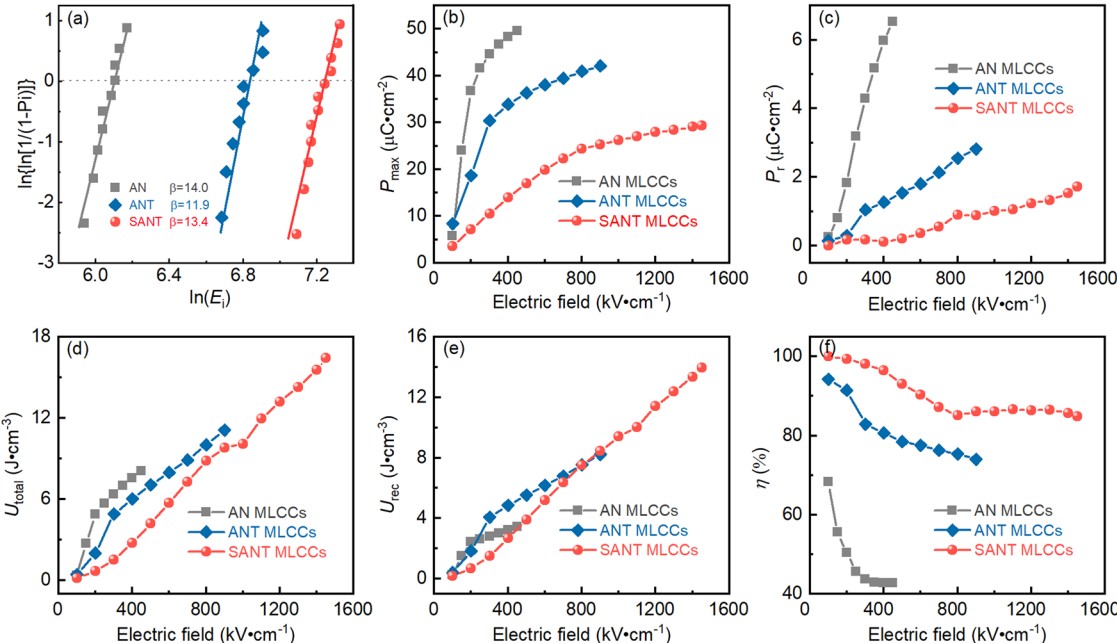

**Fig. 5 | The BDS, and ferroelectric and energy storage performances of AN, ANT, and SANT samples.** Weibull plots of DBS (**a**), the detailed variations of $P_{max}$ (**b**), $P_r$ (**c**), $U_{total}$ (**d**), $U_{rec}$ (**e**), and $\eta$ values (**f**) with different measured electric fields for AN, ANT, and SANT MLCCs.

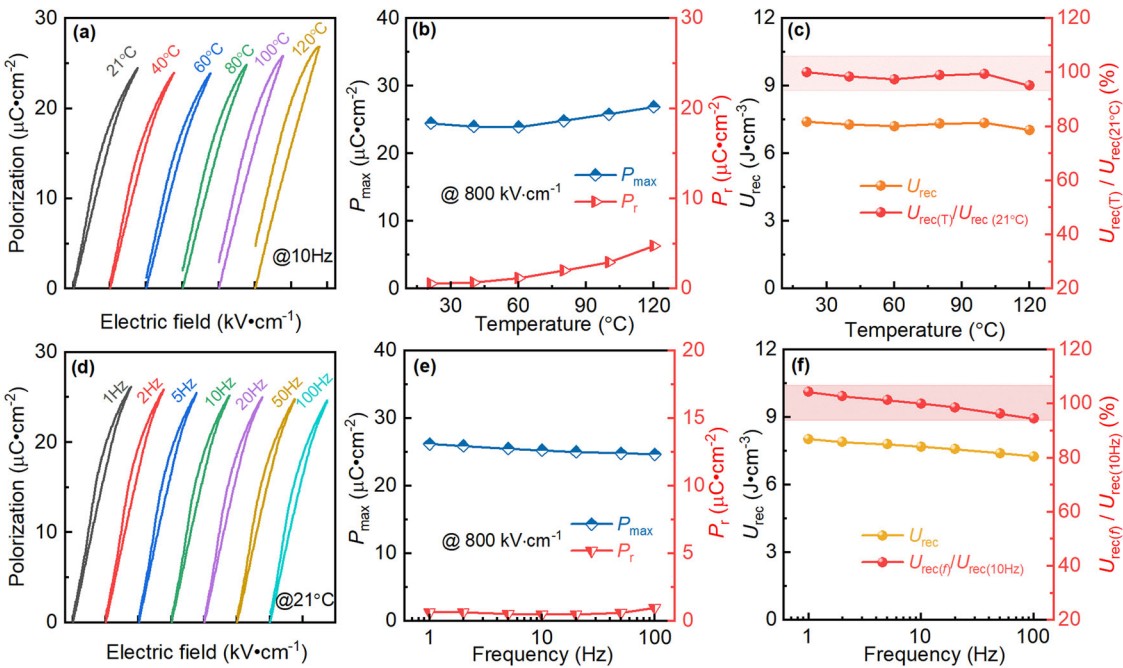

**Fig. 6 | Reliability of energy storage performance under various conditions for the SANT MLCCs.** Temperature dependence of unipolar P–E loops (**a**), $P_{max}$ and $P_r$ (**b**), $U_{rec}$ and $U_{rec}$ (T)/$U_{rec}$ (21 °C) (**c**) measured at 800 kV cm⁻¹ and 10 Hz, and the frequency dependence of unipolar P–E loops (**d**), $P_{max}$ and $P_r$ (**e**), $U_{rec}$ and $U_{rec}$ (**f**)/$U_{rec}$ (10 Hz) (**f**) measured at 800 kV cm⁻¹ and 21 °C for SANT MLCCs.

frequencies between 1 and 100 Hz is also less than 5%, as shown in Fig. 6f. These results indicate that SANT MLCCs possess excellent temperature and frequency stability, and suggest good application prospects in pulsed-discharge and power conditioning electronic devices.

The charge–discharge behavior of SANT MLCCs is also investigated to evaluate the practical application performance. The electric-field-dependent (200–1400 kV cm⁻¹) overdamped discharge current curves for SANT MLCCs at room temperature are exhibited in Fig. 7a. It can be seen that the current rapidly comes up to the maximum value

with the overall discharge duration lasting less than 1 μs. Fig. 7b summarizes the variations of electric-field dependent $P_{D,max}$ (power density $P_{D,max} = E \cdot I_{max}/2S$; $S$ is the electrode area) and $I_{max}$ (maximum discharge current), where the $P_{D,max}$ and $I_{max}$ monotonically increase as the electric field increases. Ultrahigh $P_{D,max}$ ~102.84 MW cm⁻³ and $I_{max}$ ~13.6 A can be achieved in SANT MLCCs when the electric field reaches 1400 kV cm⁻¹. As shown in Fig. 7b, a noticeable jump for electric field-dependent $P_{D,max}$ and $I_{max}$ can be seen in the electric field range of 700–900 kV cm⁻¹, which originates from the AFE to FE phase transition at high electric fields. Fig. 7c plots the room temperature

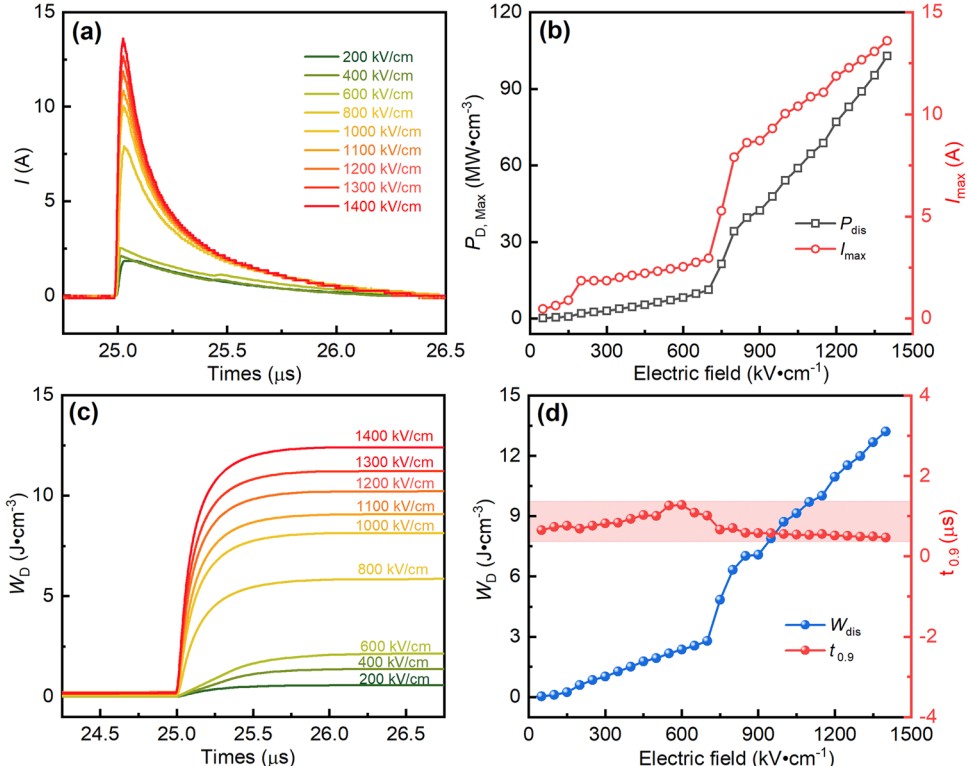

**Fig. 7 | Charge/discharge performance of SANT MLCCs.** Overdamped discharge current wave of SANT MLCCs at different electric fields (**a**), the evolution of $P_{D, Max}$ and $I_{max}$ versus the electric field (**b**), $W_D$ as a function of time ($R = 400\,\Omega$) (**c**), $W_D$ and $t_{0.9}$ values versus the electric field ($R = 400\,\Omega$) (**d**).

overdamped discharge–current–density curves of SANT MLCCs under different electric fields. The discharge-current-density rapidly reaches its maximum value in less than 0.2 µs. The discharge-energy-density ($W_d$) values can be calculated according to the current–time waveforms based on the following formula[34]:

$$W_D = R\frac{\int I(t)^2 dt}{V} \qquad (1)$$

where $R$, $I$, $t$, $V$ are the load resistor (400 $\Omega$), current, time, and sample volume, respectively. The $W_D$ shows an increase as the electric field increases. The detailed variations of $W_D$ and $t_{0.9}$ are shown in Fig.7d. The $W_D$ increases from 0.6 to 13.2 J cm$^{-3}$, with the electric fields increasing from 200 to 1400 kV cm$^{-1}$. The discharged–energy–density value calculated by the charge-discharge method ($W_D = 13.2$ J cm$^{-3}$) and by the integration of P–E loops ($U_{rec} = 13.34$ J cm$^{-3}$) are almost identical when the measured electric field is at 1400 kV cm$^{-1}$. This result suggests that its discharged–energy–density value for SANT MLCCs is credible. Different from increasing $W_D$ with increasing the electric field, the parameter $t_{0.9}$, which represents the discharge time required to release 90% of charged energy density, shows small fluctuation in the entire tested field range, which is less than 1 µs as shown in Fig. 7d. Overall, the high $P_{D,max}$, ultrahigh $W_D$, and ultrafast $t_{0.9}$ demonstrate that the SANT MLCCs have a great potential application in advanced high power/pulsed pulse systems.

In summary, high energy storage density ($U_{rec}$ ~14 J cm$^{-3}$) and efficiency ($\eta$ ~85%) are achieved simultaneously in SANT MLCCs by judiciously designing the "low-atom-displacements" region in the dielectric material, which is beneficial to reduce the electric-field-induced internal stress of MLCCs and improve the BDS. SANT MLCCs exhibit a broad usage temperature range of up to 120 °C, with minimal variations of less than 5 % for energy storage density. Meanwhile, the minimal variations in energy storage density and efficiency as functions of frequency reveal excellent frequency stability. All three merits

suggest that SANT MLCCs have a good application prospect in pulsed discharge and power conditioning electronic devices. This work proposed a new avenue for AFE MLCCs to achieve ultrahigh comprehensive energy storage performance, which meets the urgent demand for advanced high-power or pulsed power capacitors.

## Methods
### MLCCs fabrication
Silver oxide (Ag$_2$O, 99.7%), Niobium pentoxide (Nb$_2$O$_5$, 99.99%), tantalum oxide (Ta$_2$O$_5$, 99.99%), samarium oxide (Sm$_2$O$_3$, 99.9%), and manganese dioxide (MnO$_2$, 99.0%) were adopted as raw materials. These powders were weighed as following the compositions: AgNbO$_3$ (AN), Ag(Nb$_{0.70}$Ta$_{0.30}$)O$_3$ (ANT) and (Sm$_{0.05}$Ag$_{0.85}$)(Nb$_{0.70}$Ta$_{0.30}$)O$_3$ (SANT), and mixed in ethanol. The mixed powders were dried, pressed, and then calcined at 900 °C for 5 h in the oxygen atmosphere. The calcined bulks were crushed and milled for the next processing. The AN, ANT, and SANT MLCCs samples were prepared by the tape-casting and cofiring method. In order to achieve a suitable slurry, the milled AN, ANT, and SANT precursor powders were weighed, respectively, whose weight is 100 wt%, and were mixed with a solution of ethanol (21.68 wt%), Butanone (32.52 wt%), menhaden fish oil (1.01 wt%), polyethylene glycol 400 (1.87 wt%), butyl benzyl phthalate (1.87 wt%) and Polyvinyl Butyral (PVB, 6.68 wt%), as well as 0.25 wt% MnO$_2$.

These compounds were milled for 24 h and mixed well. The laboratory-type tape-casting machine with a doctor blade casting head (100 µm opening) was used for tape casting, and 75 µm thick silicone-coated mylar (polyethylene terephthalate) was also adopted as a carrier film. The green tapes, whose thickness was about 25–40 µm were cut into square samples of 25 mm in length for lamination. Pt paste was screen-printed as an internal electrode on top of the dielectric layer with a thickness of 2–3 µm. Screen-printed layers were stacked and laminated to form AN, ANT, and SANT MLCCs using a uniaxial hot press at 70 °C.

## Ferroelectric measurements

Ferroelectric hysteresis (P–E) loops were tested using an RT6000HVA ferroelectric tester (Radiant Technologies, Inc., Albuquerque, NM).

## Dielectric measurements

The curves of the relative dielectric constant versus the temperature were tested using a programmable furnace with an LCR analyzer (TH2828S) at different frequencies in the temperature range of 20–515 °C.

## Charge–discharge measurements

The actual discharge capacity was measured using a CFD-003 charge–discharge test system (Tongguo Technology, China).

## Characterization of phase and microstructure

The microstructure of the multilayer capacitors was observed by using field-emission scanning electron microscopy (FESEM, SUPRATM 55, Japan) in combination with energy dispersive spectroscopy (EDS). The crystal structure was investigated by using X-ray diffraction (XRD, Panalytical Empyrean) with Cu $K_a$ radiation ($\lambda = 1.5406$ Å) filtered through a Ni foil. A spherical aberration-corrected Titan Themis transmission electron with a double tilting stage, operating at 300 kV, was adopted to investigate the atomic-scale scanning transmission electron microscopy (STEM) of AN, ANT, and SANT samples. The detector is a high-angle annular dark field (HAADF) detector, the camera length is 115 mm, as well as the corresponding collection semi-angle range, is 48–200 mrad.

## Data availability

All data supporting this study and its findings are available within the article and its Supplementary Information. Any data deemed relevant are available from the corresponding author upon request.

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

## Acknowledgements

This work was financially supported by the National Natural Science Foundation of China (No. 52272104, and 52032007), Guangdong Basic and Applied Basic Research Foundation (No. 2019A1515110688), State Key Laboratory of New Ceramic and Fine Processing Tsinghua University (No. KF202113), and Interdisciplinary Research Project for Young Teachers of USTB (Fundamental Research Funds for the Central Universities) (No. FRF-IDRY-21-002). This work made use of the resources of the Beijing National Center for Electron Microscopy at Tsinghua University.

## Author contributions

The work was conceived and designed by L.F.Z., Y.Y., and L.Z. L.F.Z., L.Z., and L.L. fabricated the samples, tested the energy storage, dielectric, structure, stability, and other properties, and processed related data, assisted by Q.W. and H.Q. The HAADF-STEM images were filmed and processed by S.D. and G.L. The paper was drafted by L.F.Z., and revised by B.P.Z., J.C., and J.F.L. All authors participated in the data analysis and discussions.

## Competing interests

The authors declare no competing interests.
