## [Peer Review File · Nature Communications]

Heterovalent-doping-enabled atom-displacement fluctuation leads to ultrahigh energy-storage density in AgNbO₃-based multilayer capacitorsREVIEWER COMMENTS

Reviewer #1 (Remarks to the Author):

A high energy storage density ($U_{rec} \sim 14 \text{ J} \cdot \text{cm}^{-3}$ and efficiency ($\eta \sim 85\%$) are achieved in SANT MLCCs via modified phase structure and improved BDS. Authors tried to establish the physical relations between the high energy storage performance and low-atom-displacements (LAD) regions. Besides, SANT MLCCs exhibits broad usage temperature range up to $120 \text{ }^\circ\text{C}$, with minimal variations less than 5 % for energy storage density. Those results of SANT MLCCs manifest itself that they can exhibit a good application prospect in pulsed-discharge and power conditioning electronic devices. The high energy storage performance here is indeed attractive and important for applications of AN based materials in electronic industry, however, the reasons for high energy storage performance of SANT and physical explanation here is not very satisfied. Therefore, major revision is suggested before it can be accepted for publication to Nature Comms standard. Reasons for my opinion are:

According to literature review part, MLCCs technique has been proved to be the most effective strategy to boost the BDS and U_{rec} of the AgNbO_3 (AN) system. Authors also have reported ultrahigh BDS = $1020 \text{ kV} \cdot \text{cm}^{-1}$, excellent $U_{rec} = 7.9 \text{ Jcm}^{-3}$ and $\eta = 71\%$ in $\text{Ag}(\text{Nb}_{0.85}\text{Ta}_{0.15})\text{O}_3$ (ANT) MLCC. Besides, according to authors' idea in Figure 1, the heterovalent-doping-induced low-atom-displacements (LAD) regions in the $\text{Sm}_{0.05}\text{Ag}_{0.85}\text{Nb}_{0.7}\text{Ta}_{0.3}\text{O}_3$ (SANT) antiferroelectric matrix was responsible for the increase in BDS and enhancement of the energy-storage density. Therefore, based on this logic, the dominant contribution of ultrahigh performance in SANT is from stage II of appearance of LAD region, rather than stage I of phase transition, which occurred in ANT compared with AN (fig.1f). Therefore, the evidence and physical explanations the authors offered in the article should focus on stage II of appearance of LAD region, rather than stage I of phase transition. That is to say, in figure1(e), figure 2, figure S1, the corresponding data of SANT and ANT samples should be compared, at least supplemented, to clarify why LAD regions in SANT MLCCs give rise energy storage performance superior to ANT MLCCs? Why comes the ultrahigh BDS while decreased P_{max} and P_r in 4 and 5? The physical implication under SANT and ANT samples is very necessary and important for the novelty of this work.

In addition, in figure S1, the parameter of different electrical fields for different samples should be presented

Some minor errors in the manuscript are also listed:

- (1) Line 26, the formula " $\text{Sm}_{0.05}\text{Ag}_{0.85}\text{Nb}_{0.7}\text{Ta}_{0.3}$ " should be " $\text{Sm}_{0.05}\text{Ag}_{0.85}\text{Nb}_{0.7}\text{Ta}_{0.3}\text{O}_3$ ".
- (2) Line 43 and line 51, remanent polarization (P_r) should appeared first in Line 43, rather than in line 51.
- (3) Line 321, a space between "Li" and "fabricated" is needed.
- (4) Captions of Fig. S3-S6, "at different electric fields" rather than "at different electric field".

Reviewer #2 (Remarks to the Author):

This manuscript entitled 'Heterovalent-doping-enabled atom-displacement fluctuation leads to ultrahigh 2 energy-storage density in AgNbO_3 -based multilayer capacitors' reports a heterovalent doping atom displacement for the antiferroelectric AgNbO_3 -based MLCCs to achieve the maximum energy density. The MLCC shows an outstanding breakdown strength

(BDS) of 1450 kV/cm as well as storage energy density of 14 J/cm³. This manuscript is well written with scientific experimental process and the result is also good, thus enable to accept in Nature Communication after addressing below comments.

1. The authors claimed that the MLCC system has a higher BDS and storage energy density compared to the ceramic bulks. What is the reason that MLCC shows such outstanding performance compared to the bulks?
2. In Figure 7, the discharge-energy-density value of SANT MLCC is noticeably jumped between 700 to 900 kV/cm. Why the value is instantly increased?

Reviewer #3 (Remarks to the Author):

The present manuscript reported that high energy storage density (14 J·cm⁻³) and efficiency (85%) can be realized by proposing a heterovalent-doping-enabled atom-displacement fluctuation strategy for the design of low-atom-displacements (LAD) regions in Sm_{0.05}Ag_{0.85}Nb_{0.7}Ta_{0.3}O₃ MLCCs, which are supported by high resolution HAADF images. However, I am not well convinced that the paper deserves publication in Nat. Communi. from the perspectives of property and approach. There are several works reported the excellent energy storage performance of MLCCs, such as Wrec of 14.49 J·cm⁻³ and η of 84.9% in 0.4((Bi_{0.5}Na_{0.5})TiO₃)-0.6(0.87BaTiO₃-0.13Bi(Zn_{2/3}(Nb_{0.85}Ta_{0.15})_{1/3})O₃) ceramics (J. Mater. Chem. A, 2021, 9, 25914), Wrec of 18.24 J·cm⁻³ and η of 94.5% in 0.87BaTiO₃-0.13Bi(Zn_{2/3}(Nb_{0.85}Ta_{0.15})_{1/3})O₃ ceramics (Energy Environ. Sci., 2020, 13, 4882), Wrec of 18 J·cm⁻³ in NBT-SBT-0.08BMN ceramics (Energy Storage Materials 2021, 38, 113-120), Wrec of 21.5 J·cm⁻³ in 0.65BNT-0.35SBT textured ceramics (Nature Materials, 19, 2020, 999-1005) and so on. The above material systems referred to BT/BNT-based, which possess much lower cost than AN-based ceramics in this work. In addition, I have the following points that need author's attention.

1. In the abstract, Sm_{0.05}Ag_{0.85}Nb_{0.7}Ta_{0.3} should be Sm_{0.05}Ag_{0.85}Nb_{0.7}Ta_{0.3}O₃.
2. The dielectric curves of ANT and SANT specimens have no obvious difference, while their P-E loops show large discrepancy. What's the role of the Sm? The HAADF image of ANT is lacking, it's insufficient to argue that LAD region can't form in ANT system.
3. The key point/novelty of the present work could be atom-displacement fluctuation, is there any other characterization method to supplement the HAADF image?
4. It's known that the organic dielectric thin films have been widely investigated in the field of energy storage recently. Can the author give some comments on the advantages of SANT MLCCs for energy storage applications in compared with the organic dielectric thin films?

Dear Editor

We would like to thank you very much for your valuable reviewing of our manuscript (NCOMMS-22-51392) and provoking suggestions. Following is our answers to the referee's comments, according to which we have thoroughly revised our manuscript. We hope the revised manuscript will be suitable for the publication in your *Nature Communications*.

Responses to specific comments raised by the reviewers

Report of Reviewer 1

A high energy storage density ($U_{\text{rec}} \sim 14 \text{ J}\cdot\text{cm}^{-3}$ and efficiency ($\eta \sim 85\%$) are achieved in SANT MLCCs via modified phase structure and improved BDS. Authors tried to establish the physical relations between the high energy storage performance and low-atom-displacements (LAD) regions. Besides, SANT MLCCs exhibits broad usage temperature range up to 120 °C, with minimal variations less than 5 % for energy storage density. Those results of SANT MLCCs manifest itself that they can exhibit a good application prospect in pulsed-discharge and power conditioning electronic devices. The high energy storage performance here is indeed attractive and important for applications of AN based materials in electronic industry, however, the reasons for high energy storage performance of SANT and physical explanation here is not very satisfied. Therefore, major revision is suggested before it can be accepted for publication to Nature Comms standard. Reasons for my opinion are:.

We sincerely thank the reviewer for his/her positive comments and for recommending our manuscript for publication in Nature Comms. Indeed, the extensive SI that contains many detailed and interesting theoretical results also support the conclusions of our work.

Comment #1: According to literature review part, MLCCs technique has been proved to be the most effective strategy to boost the BDS and U_{rec} of the AgNbO_3 (AN) system. Authors also have reported ultrahigh BDS = $1020 \text{ kV}\cdot\text{cm}^{-1}$, excellent $U_{\text{rec}} = 7.9 \text{ Jcm}^{-3}$ and $\eta = 71\%$ in $\text{Ag}(\text{Nb}_{0.85}\text{Ta}_{0.15})\text{O}_3$ (ANT) MLCC. Besides, according to authors' idea in Figure 1, the heterovalent-doping-induced low-atom-displacements (LAD) regions in the $\text{Sm}_{0.05}\text{Ag}_{0.85}\text{Nb}_{0.7}\text{Ta}_{0.3}\text{O}_3$ (SANT) antiferroelectric matrix was responsible for the increase in BDS and enhancement of the energy-storage density. Therefore, based on this logic, the dominant contribution of ultrahigh performance in SANT is from stage II of appearance of LAD region, rather than stage I of phase transition, which occurred in ANT compared with AN (fig.1f). Therefore, the evidence and physical explanations the authors offered in the article should focus on stage II of appearance

of LAD region, rather than stage I of phase transition. That is to say, in figure1(e), figure 2, figure S1, the corresponding data of SANT and ANT samples should be compared, at least supplemented, to clarify why LAD regions in SANT MLCCs give rise energy storage performance superior to ANT MLCCs?

Response: Thank you very much for your good suggestions. Following the reviewer's suggestions, we have added additional information about the ANT sample and compared with those of AN and SANT to solidify our arguments. (1) The strain data of ANT samples has been added and compared in the revised Figure1(e). (2) Atomically resolved HAADF-STEM images of ANT sample have been analyzed and added in Figure S3, where the LAD regions are absent. The cation displacements for SANT sample possess the periodic wave along the [001]C direction, sharing the analogous characteristics with that of AN sample. Meanwhile, we note that the average peak amplitude of the waveform for ANT system (~10 pm) is slightly smaller than that of AN (~13 pm), which should originate from the lower electronegativity of Ta⁵⁺ than Nb⁵⁺. (3) The finite-element simulations for the strain distribution of ANT samples have also been added and compared in the revised Figure S2. These results double confirm the critical roles of heterovalent-doping and the induced LAD regions in improving the energy-storage performance.

Revisions: in MS Page 6, the strain data of ANT samples has been added and compared in the revised Figure1(e):

Figure 1. crystal structure of pure AN corresponding to ferrielectric Pmc₂₁ phase whose cations show displacement along the $\pm[1\bar{1}0]_c$ direction, forming a periodic variation along the c axis direction (a), crystal structure for (RE_xAg_{1-3x})(Nb,Ta)O₃

system, the periodic variation of atom-displacement fluctuation in AFE Pbcm phase was destroyed by the heterogeneous rare earth ion RE^{3+} ions, forming some low-atom-displacements (LAD) region (b). The diagram of phase transition and volume expansion process of AN (c) and $(RE_xAg_{1-3x})(Nb,Ta)O_3$ (d) capacitors as a high electric field $E > E_F$ is applied. Electric-field-induced strain of AN, ANT and SANT ceramics (e). The comparison of BDS and U_{rec} for AN-based ceramics, AN MLCCs and SANT MLCCs (f).

Revisions: in SI Page 3, simulations for the strain distribution of ANT samples have been added:

Fig.S2 MLCCs diagram (a), single layer of MLCCs (b), finite-element simulations for the strain distribution of AN-based, ANT-based and SANT-based MLCCs at different external electric fields (E). von Miss stress distribution for AN MLCCs (N/m^2) at $E < E_F$ (c) and $E > E_F$ (d), ANT-based MLCCs $E < E_F$ (e) and $E > E_F$ (f), SANT-based MLCCs at $E < E_F$ (e) and $E > E_F$ (f)

Revisions: in SI Page 5, a new supplementary figure has been added:

Fig.S3 (a) HAADF-STEM image of $\text{Ag}(\text{Nb}_{0.85}\text{Ta}_{0.15})\text{O}_3$ (ANT_{0.15}), (b) Map of the atomic displacements of Ag, Nb and Ta atoms along the $[1\bar{1}0]_c$ direction, showing a good c-axial periodicity. (c) Average displacement of each vertical atomic

plane

Comment #2: Why comes the ultrahigh BDS while decreased P_{\max} and P_r in 4 and 5?

Response: It is well recognized that the magnitudes of P_{\max} and P_r values are associated with the domain morphology and configuration. Herein, with the emergence of LAD regions, the domain morphology of SANT system has been changed in comparison to AN and ANT capacitors. Especially, it can be seen in Figure 2(e) and (f) that the cation displacements in the LAD regions are decreased, which should contribute to the decrease in P_{\max} and P_r , as observed in Figure 4 and Figure 5. Meanwhile, the emergence of LAD region is beneficial to reduce the electric-field-induced strain and electric-field-induced internal stress of MLCCs as shown in the Figure 1 and Figure S2, which are in favor of the improvement of BDS. Thus, the BDS of SANT MLCCs has been improved significantly while its P_{\max} and P_r are decreased.

Comment #3: The physical implication under SANT and ANT samples is very necessary and important for the novelty of this work.

Response: Thank you for your suggestions. Accordingly, we have added additional discussions regarding the underlying physical mechanisms.

The replacement Nb^{5+} ions by Ta^{5+} ions belongs to isovalent ion substitution, and the defect is free in the ANT samples. Figure S1a shows unit cell of $\text{Ag}(\text{Nb}_{1-x}\text{Ta}_x)\text{O}_3$ system in which the Nb^{5+} ions were replaced by Ta^{5+} ions, and two-dimensional planar plan for cationic periodic variation along the c axis direction. Because of the replacement Nb^{5+} by Ta^{5+} ions, the phase structure of $\text{Ag}(\text{Nb}_{0.7}\text{Ta}_{0.3})\text{O}_3$ capacitors has turned into an AFE phase, in which both its P_S^+ and P_S^- are equal and their direction is opposite as shown in Fig.S1b. Fig.S1c has simulated the arrangement of domains in grains of the $\text{Ag}(\text{Nb}_{0.7}\text{Ta}_{0.3})\text{O}_3$ AFE capacitors. When an electric field (E , $E > E_F$) is applied in the $\text{Ag}(\text{Nb}_{0.7}\text{Ta}_{0.3})\text{O}_3$ AFE capacitors, a phase transition from AFE to FE happens, along with a large volume expansion and electric-field-induced strain as shown in Fig.S1d.

However, for the SANT samples, the substitution Sm^{3+} for Ag^+ belongs to heterovalent ion substitution, and generates two $V_{\text{Ag}^+}^{\prime}$ defects or one $V_{\text{Ag}^+}^{\prime} - \text{RE}^{3+} - V_{\text{Ag}^+}^{\prime}$ defect dipole. To maintain the stability of unit cell structure and electric neutrality, partial Ag^+ and Nb^{5+} ions will migrate to adjacent t regions, forming LAD region as shown in Fig.1b.

Revisions: in SI Page 1, a new supplementary figure (Figure S1) has been added:

Fig.S1 crystal structure for ANT (a) corresponding to antiferroelectric Pbcm phase cations show displacement along the $\pm[1\bar{1}0]_c$ direction, forming a periodic variation along the c axis direction (b). Diagram of phase transition and volume expansion process of ANT from AFE to FE phase before (c) and after (d) applying electric field $E > E_F$.

Revisions: in SI Page 1, the relative mechanisms about the structure and strain for ANT samples have been added.

Figure S1a shows unit cell of $\text{Ag}(\text{Nb}_{1-x}\text{Ta}_x)\text{O}_3$ system in which the Nb^{5+} ions were replaced by Ta^{5+} ions, and two-dimensional planar plan for cationic periodic variation along the c axis direction. Because of the replacement Nb^{5+} by Ta^{5+} ions, the phase structure of $\text{Ag}(\text{Nb}_{0.7}\text{Ta}_{0.3})\text{O}_3$ capacitors has turned into an AFE phase, in which both its P_s^+ and P_s^- are equal and their direction is opposite as shown in Fig.S1b. Fig.S1c has simulated the arrangement of domains in grains of the $\text{Ag}(\text{Nb}_{0.7}\text{Ta}_{0.3})\text{O}_3$ AFE capacitors. When an electric field (E , $E > E_F$) is applied in the $\text{Ag}(\text{Nb}_{0.7}\text{Ta}_{0.3})\text{O}_3$ AFE capacitors, a phase transition from AFE to FE happens, along with a large volume expansion and electric-field-induced strain as shown in Fig.S1d.

Comment #4: In addition, in figure S1, the parameter of different electrical fields for different samples should be presented

Response: Following the reviewer’s suggestions, the parameters for finite-element simulations for AN-based, ANT, and SANT MLCCs samples have been added in Table S1.

Revisions: in SI Page 3, a new supplementary table has been added:

Table S1 the parameter of finite-element simulations for AN-based, ANT and SANT MLCCs

Materials	E	Flexibility matrix						Relative dielectric constant		
		S_{11}	S_{12}	S_{13}	S_{33}	S_{44}	S_{66}	ϵ_{11}	ϵ_{22}	ϵ_{33}
AN	$E < E_F$	1.17	-0.335	-0.537	1.52	3.04	2.97	320	320	305
	$E > E_F$	1.25	-0.358	-0.573	1.62	3.25	3.17	430	430	415
ANT	$E < E_F$	1.18	-0.338	-0.532	1.51	3.02	2.94	430	430	410
	$E > E_F$	1.24	-0.355	-0.568	1.61	3.22	3.14	405	405	380
SANT	$E < E_F$	1.20	-0.344	-0.550	1.56	3.12	3.04	445	445	420
	$E > E_F$	1.23	-0.353	-0.564	1.59	3.20	3.12	420	420	405

Comment #5: Some minor errors in the manuscript are also listed:

- (1) Line 26, the formula “Sm0.05Ag0.85Nb0.7Ta0.3” should be “Sm0.05Ag0.85Nb0.7Ta0.3O3”.
- (2) Line 43 and line 51, remanent polarization (P_r) should appeared first in Line 43, rather than in line 51.
- (3) Line 321, a space between “Li” and “fabricated” is needed.
- (4) Captions of Fig. S3-S6, “at different electric fields” rather than “at different electric field”.

Response: Response: Thank you very much for your suggestions. We have made revisions accordingly.

- (1) The formula “Sm0.05Ag0.85Nb0.7Ta0.3” has been changed to “Sm0.05Ag0.85Nb0.7Ta0.3O3” in the revised manuscript.
- (2) The “ P_r ” in the line 43 has been changed to the “remanent polarization (P_r)”, and the “remanent polarization (P_r)” in the line 51 was changed to the “ P_r ”.
- (3) A space between “Li” and “fabricated” has been added in the line 321.
- (4) The words of “at different electric field” has been changed to “at different electric fields” in the Captions of Fig. S3-S6.

Revisions: MS Page 5,

Report of Reviewer 2

This manuscript entitled ‘Heterovalent-doping-enabled atom-displacement fluctuation leads to ultrahigh 2 energy-storage density in AgNbO₃-based multilayer capacitors’ reports a heterovalent doping atom displacement for the antiferroelectric AgNbO₃-based MLCCs to achieve the maximum energy density. The MLCC shows an outstanding breakdown strength (BDS) of 1450 kV·cm⁻¹ as well as storage energy density of 14 J·cm⁻³. This manuscript is well written with scientific experimental process and the result is also good, thus enable to accept in Nature Communication after addressing below comments.

We sincerely thank the reviewer for your positive comments and for recommending our manuscript for publication in Nature Communications.

Comment #1: The authors claimed that the MLCC system has a higher BDS and storage energy density compared to the ceramic bulks. What is the reason that MLCC shows such outstanding performance compared to the bulks?

Response: There are two reasons for the outstanding BDS and storage energy density achieved our MLCC system. **One** reason is the thickness. The thickness of each dielectric layers in MLCC is as low as about 10 μm, which is far lower than that of ceramic bulk (~200 μm). The low thickness is beneficial for the improvement of BDS because of $BDS \propto (\text{thickness})^{-\alpha}$ ($\alpha \sim 0.5$), as reported by Yang et al (Progress in Materials Science 102, 72-108, 2019). As a result, the storage energy density can be enhanced in MLCC. **The other** reason is the grain size. The homogeneous and thin grain size can be achieved easily in the MLCCs, which can also benefit the improvement of BDS and storage energy density.

Comment #2: In Figure 7, the discharge-energy-density value of SANT MLCC is noticeably jumped between 700 to 900 kV·cm⁻¹. Why the value is instantly increased?

Response: Thank you very much for your good suggestions. The noticeable jump of the discharge-energy-density value for the SANT MLCC at the electric field between 700 to 900 kV·cm⁻¹ comes from the AFE to FE phase transition at the high electric fields. This can be confirmed by variations of P_{\max} (a) and U_{total} (b) with different measured electric fields for AN, ANT, and SANT MLCCs, as shown in following Figure R1. As demonstrated in Figure R1(a), two differential-slope-coefficient lines that correspond to the l1 and l2 can

be detected, and their intersection lies in the electric-field interval between 700 to 900 $\text{kV} \cdot \text{cm}^{-1}$. It is well known that the phase structure of SANT system is the AFE phase at low electric fields and turns into the FE phase at high electric fields. Thus, such a phase transition should be a AFE to FE transition. Besides, Figure R1(b) shows additional evidence, where the total energy storage density has an obvious abrupt change as the electric field lies the interval between 700 to 900 $\text{kV} \cdot \text{cm}^{-1}$. Following the reviewer's suggestions, we have added additional discussions in our revised manuscript.

Revisions: in MS Page 14, line 12, additional discussions have been added:

As shown in Fig. 7b, a noticeable jump for electric field-dependent $P_{D,\max}$ and I_{\max} can be seen in the electric field range of 700 to 900 $\text{kV} \cdot \text{cm}^{-1}$, which originates from the AFE to FE phase transition at high electric fields.

Figure R1 the detailed variations of P_{\max} (a) and U_{total} (b) with different measured electric fields for AN, ANT, and SANT MLCCs

Report of Reviewer 3

The present manuscript reported that high energy storage density ($14 \text{ J} \cdot \text{cm}^{-3}$) and efficiency (85%) can be realized by proposing a heterovalent-doping-enabled atom-displacement fluctuation strategy for the design of low-atom-displacements (LAD) regions in $\text{Sm}_{0.05}\text{Ag}_{0.85}\text{Nb}_{0.7}\text{Ta}_{0.3}\text{O}_3$ MLCCs, which are supported by high resolution HAADF images. However, I am not well convinced that the paper deserves publication in Nat. Communi. from the perspectives of property and approach. There are several works reported the excellent

energy storage performance of MLCCs, such as W_{rec} of $14.49 \text{ J}\cdot\text{cm}^{-3}$ and of 84.9% in $0.4(\text{Bi}_{0.5}\text{Na}_{0.5})\text{TiO}_3$ - $0.6(0.87\text{BaTiO}_3$ - $0.13\text{Bi}(\text{Zn}_{2/3}(\text{Nb}_{0.85}\text{Ta}_{0.15})_{1/3})\text{O}_3)$ ceramics (J. Mater. Chem. A, 2021, 9, 25914), W_{rec} of $18.24 \text{ J}\cdot\text{cm}^{-3}$ and of 94.5% in 0.87BaTiO_3 - $0.13\text{Bi}(\text{Zn}_{2/3}(\text{Nb}_{0.85}\text{Ta}_{0.15})_{1/3})\text{O}_3$ ceramics (Energy Environ. Sci., 2020, 13, 4882), W_{rec} of $18 \text{ J}\cdot\text{cm}^{-3}$ in NBT-SBT-0.08BMN ceramics (Energy Storage Materials 2021, 38, 113-120), W_{rec} of $21.5 \text{ J}\cdot\text{cm}^{-3}$ in 0.65BNT-0.35SBT textured ceramics (Nature Materials, 19, 2020, 999-1005) and so on. The above material systems referred to BT/BNT-based, which possess much lower cost than AN-based ceramics in this work. In addition, I have the following points that need author's attention.

Response: Thank you for the comments. We would like to emphasize that the novelty and significance of this work lie in the following three aspects.

(1) One is the ability to greatly improve the comprehensive energy storage performance of the prototypical antiferroelectric AgNbO_3 system. The achieved BDS ($\sim 1450 \text{ kV}\cdot\text{cm}^{-1}$) and energy-storage density ($\sim 14 \text{ J}\cdot\text{cm}^{-3}$) of SANT MLCCs in our work are both the highest values in AgNbO_3 antiferroelectrics-based systems so far.

(2) Our developed new strategy of the heterovalent-doping using rare earth ion RE^{3+} ions can be generally applicable to numerous antiferroelectrics, as well as ferroelectrics and dielectrics, which would navigate the discovery and development of superior energy storage materials.

(3) The newly discovered mechanism for the energy storage performance improvement, that is, appearance of LAD region, is markedly different from the traditional ones, such as the typical one of polar nano-regions (PNRs) in relaxor ferroelectrics. This would greatly advance our understanding the nature of the complicated local structures for superior energy storage performance especially for different types of materials. New thinking on this emergent mechanism can provide foundation for the development of diverse methodologies for the design of high-performance energy storage materials.

Comment #1: In the abstract, $\text{Sm}_{0.05}\text{Ag}_{0.85}\text{Nb}_{0.7}\text{Ta}_{0.3}$ should be $\text{Sm}_{0.05}\text{Ag}_{0.85}\text{Nb}_{0.7}\text{Ta}_{0.3}\text{O}_3$..

Response: Thank you very much for pointing out this careless mistake. We have made corresponding changes in our revised manuscript.

Revisions: MS Page 2, Abstract, line 3:

An ultrahigh BDS $\sim 1450 \text{ kV}\cdot\text{cm}^{-1}$ is realized in the $\text{Sm}_{0.05}\text{Ag}_{0.85}\text{Nb}_{0.7}\text{Ta}_{0.3}\text{O}_3$ MLCCs, especially with an ultrahigh $U_{\text{rec}} \sim 14 \text{ J}\cdot\text{cm}^{-3}$, excellent $\eta \sim 85\%$ and $P_{\text{D,max}} \sim 102.84 \text{ MW}\cdot\text{cm}^{-3}$.

Comment #2.1: The dielectric curves of ANT and SANT specimens have no obvious difference, while their P-E loops show large discrepancy. What's the role of the Sm?

Response: Thank you for the questions. Since the dielectric curves reflect the effect of temperature on the dielectric constant, while P-E loops address the effect of electric field on polarization, doping Sm^{3+} ions in ANT should mainly influence the local electronic structures to form defect dipoles to modulate the polarization property. Specifically, the critical roles of the earth Sm^{3+} ions are illustrated in Figure 1 and the newly added Figure S1. The replacement Ag^+ by Sm^{3+} can produce two $V_{\text{Ag}^+}^{\cdot}$ defects or one $V_{\text{Ag}^+}^{\cdot}-\text{RE}^{3+}-V_{\text{Ag}^+}^{\cdot}$ defect dipole, making the Ag^+ and Nb^{5+} ions migrate to adjacent t regions to the $V_{\text{Ag}^+}^{\cdot}$ and forming LAD region. In this sense, the crystal structure and two-dimensional planar plan for cationic periodic variation along the c-axial direction of ANT sample can be changed, resulting in the changes in the macroscopic P-E loops. Accordingly, we have added these discussions in our revised supporting materials.

Revisions: in SI Page 1, Figure S1, a new supplementary figure has been added:

Revisions: in SI Page 1, additional discussions have been added:

Fig.S1 crystal structure for ANT (a) corresponding to antiferroelectric Pbcm phase cations show displacement along the $\pm [1 \bar{1} 0]_c$ direction, forming a periodic variation along the c axis direction (b). Diagram of phase transition and volume expansion process of ANT from AFE to FE phase before (c) and after (d) applying electric field $E > E_F$.

Comment #2.2: The HAADF image of ANT is lacking, it's insufficient to argue that LAD region can't form in ANT system.?

Response: Thank you for the suggestions. Following the reviewer's suggestions, we have acquired the atomically resolved HAADF-STEM images for and conducted quantitative analysis. The results are now included in Figure S3 in the revised supporting information. As shown in Figure S3(b), the LAD regions are also absent in the ANT system. Displacements of Ag, Nb and Ta atoms along the $[1-10]_c$ direction show a good periodicity, being similar to the scenario in AN sample (Figure 2b).

Revisions: SI Page 4, Figure S3, a new supplementary figure has been added:

Fig.S3 (a) HAADF-STEM image of $\text{Ag}(\text{Nb}_{0.85}\text{Ta}_{0.15})\text{O}_3$ ($\text{ANT}_{0.15}$), (b) Map of the atomic displacements of Ag, Nb and Ta atoms along the $[1-10]_c$ direction, showing a good c-axial periodicity. (c) Average displacement of each vertical atomic

plane

Comment #3: The key point/novelty of the present work could be atom-displacement fluctuation, is there any other characterization method to supplement the HAADF image?

Response: As the atom-displacement fluctuation is randomly distributed in the sample and its size at the nanometer scale range, other characterization methods like synchrotron X-ray and neutron diffraction are difficult to directly detect these local microstructures. Even though, the influences of such atom-displacement fluctuation can be reflected by macroscopic characterizations, like P-E loops. It is well known that the P_{\max} value is relative to the domain morphology and domain alignment. For example, high P_{\max} value corresponds to a highly ordered electric dipoles in the same direction. Herein, being different from the AN and ANT samples, the SANT MLCCs have a low P_{\max} values as shown in Fig 4 and 5. This can imply that the domain morphology and domain alignment for SANT samples is different from AN and ANT samples, and the SANT MLCCs should have a relatively more disordered electric dipoles than that of AN and ANT.

Comment #4: It's known that the organic dielectric thin films have been widely investigated in the field of energy storage recently. Can the author give some comments on the advantages of SANT MLCCs for energy storage applications in compared with the organic dielectric thin films?

Response: Dielectric materials include organics (biaxially-oriented polypropylene, abbreviated as BOPP; polyetherimide, abbreviated as PEI; polyvinylidene fluoride, abbreviated as PVDF, etc.), inorganics (ceramics, glass, glass-ceramics, etc.), and their composites. However, each dielectric material seems to have its limitation. For example, polymers often possess high breakdown strength, but low dielectric constant and weak stability to thermal stimulus.

Taking many factors into account, such as energy storage potential, adaptability to multifarious environment, fundamentality and et al, ceramic-based dielectrics have already become current research focus as illustrated by soaring rise of publications associated with energy storage ceramics. SANT capacitors as antiferroelectric materials have attracted extensive attention in recent years, especially its excellent energy-storage density and temperature stability properties. Here, SANT MLCCs possess an ultrahigh $U_{\text{rec}} \sim 14 \text{ J}\cdot\text{cm}^{-3}$ and excellent $\eta \sim 85\%$ as well as high $P_{\text{D,max}} \sim 102.84 \text{ MW}\cdot\text{cm}^{-3}$, making it has a great potential in advancing energy storage applications

REVIEWERS' COMMENTS

Reviewer #1 (Remarks to the Author):

The manuscript has been revised according to my opinion, it can be accepted after typo correction in Support information.

P3, Line 37-38, "SANT-based MLCCs at $E < E_F$ (e) and $E > E_F$ (f)" should be "SANT-based MLCCs at $E < E_F$ (g) and $E > E_F$ (h)"

Reviewer #2 (Remarks to the Author):

In this revision, the authors have addressed the questions properly. In this case, it is ready to accept this work in the current format.

Dear Editor

We would like to thank you very much for your valuable reviewing of our manuscript (NCOMMS-22-51392B) and provoking suggestions. Following is our answers to the referee's comments, according to which we have thoroughly revised our manuscript. We hope the revised manuscript will be suitable for the publication in your *Nature Communications*.

Responses to specific comments raised by the reviewers

Report of Reviewer 1

The manuscript has been revised according to my opinion, it can be accepted after typo correction in Support information.

We sincerely thank the reviewer for recommending our manuscript for publication in Nature Communications.

Comment #1: P3, Line 37-38, "SANT-based MLCCs at $E < E_F$ (e) and $E > E_F$ (f)" should be "SANT-based MLCCs at $E < E_F$ (g) and $E > E_F$ (h)"

Response: Thank you very much for your suggestions. We have made revisions accordingly.

(1) The "SANT-based MLCCs at $E < E_F$ (e) and $E > E_F$ (f)" in the line 37-38 have been changed to the "SANT-based MLCCs at $E < E_F$ (g) and $E > E_F$ (h)".

Revisions: SI Page 5.

Report of Reviewer 2

In this revision, the authors have addressed the questions properly. In this case, it is ready to accept this work in the current format.

We sincerely thank the reviewer for recommending our manuscript for publication in Nature Communications.